# Enzymatic Methods for Mutation Detection in Cancer Samples and Liquid Biopsies

**DOI:** 10.3390/ijms24020923

**Published:** 2023-01-04

**Authors:** Farzaneh Darbeheshti, G. Mike Makrigiorgos

**Affiliations:** Department of Radiation Oncology, Dana Farber Cancer Institute, Brigham and Women’s Hospital, Harvard Medical School, Boston, MA 02215, USA

**Keywords:** mutation detection, low-level mutations, nucleotide mismatches, polymerase chain reaction

## Abstract

Low-level tumor somatic DNA mutations in tissue and liquid biopsies obtained from cancer patients can have profound implications for development of metastasis, prognosis, choice of treatment, follow-up, or early cancer detection. Unless detected, such low-frequency DNA alterations can misinform patient management decisions or become missed opportunities for personalized medicine. Next-generation sequencing technologies and digital-PCR can resolve low-level mutations but require access to specialized instrumentation, time, and resources. Enzymatic-based approaches to detection of low-level mutations provide a simple, straightforward, and affordable alternative to enrich and detect such alterations and is broadly available to low-resource laboratory settings. This review summarizes the traditional uses of enzymatic mutation detection and describes the latest exciting developments, potential, and applications with specific reference to the field of liquid biopsy in cancer.

## 1. Introduction

In the field of cancer, low-level tumor somatic DNA mutations can have profound implications for development of metastasis, prognosis, choice of treatment, follow-up, or early cancer detection. Unless detected, such low-frequency DNA alterations can misinform patient management decisions or become missed opportunities for personalized medicine. While next-generation sequencing (NGS) technology or digital droplet PCR (ddPCR) can identify low-level mutations, substantial time, effort, and expense are still required for NGS-based diagnostics, while ddPCR requires special instrumentation. In cases where a limited number of pre-defined, hotspot mutations are sought, or when access to advanced instrumentation is unavailable, simpler methods to remove unaltered, WT DNA alleles and enrich mutated alleles can be applied. Enzymatic approaches to eliminate WT alleles are particularly attractive in this regard due to their simplicity, low cost, and general availability in basic laboratories. This review describes traditional and recently developed enzymatic methods for mutation enrichment and detection in clinical samples, with an emphasis on detecting low-level mutations in liquid biopsies. Figure 1 schematically illustrates the three main categories of enzymatic mutation detection methods. Table 1 lists enzymatic mutation detection methods along with their sensitivity limits and intended use, towards detecting mutations at known DNA target sites or mutation scanning (discovery of unknown mutations).

## 2. Restriction Enzymes

Restriction endonucleases (REs) are conventionally employed to detect known point mutations [1,29]. They detect and cleave double-stranded DNA (dsDNA) through their specific recognition sequence with high selectivity [30]. This feature, besides their affordability, has turned REs into a commonly used approach for enriching and detecting known mutations that fall within their specific recognition site. Although their use is limited to the fraction of mutations that occur within their recognition sequence, the selectivity and sensitivity achieved for such mutations, plus their ease of use made RE-based mutation enrichment one of the first approaches to be applied in molecular diagnostics [1,29,30]. In this section, original and newer RE-based techniques for mutation detection are briefly discussed.

In 1990, Restriction-Site Mutation (RSM) detection was introduced for detecting mutations that alter RE recognition sites such that mutant DNAs show resistance to endonuclease activity [1]. Following incubation of the interrogated DNA with a specific RE, wild-type DNA is digested while mutations at any base within the enzyme recognition sequence retain DNA in intact form. A PCR that amplifies the interrogated locus with primers encompassing the recognition sequence is then performed, thereby generating products highly enriched for mutations. As an example application, RSM was applied for detection of mutations generated in mutagen-exposed tissues [1]. Following DNA extraction from tissues, mutagen-exposed DNA underwent *AluI* digestion, followed by PCR for codons 12/13 of the *N-RAS* gene. PCR amplicons were visualized on a polyacrylamide gel to resolve intact DNA indicating mutations in *N-RAS*. This early work revealed a promising method for mutation detection and propelled the development of similar RE-based mutation detection, restriction fragment length polymorphism RFLP-PCR, and PCR-RFLP [31]. RSM/RFLP methods provide semi-quantitative mutation detection while being less laborious than alternative approaches. An optimized RSM/RFLP method can detect minority mutant alleles at allelic frequencies as low as 10^−6^ mutant-to-wild type alleles. To achieve this selectivity, the RE step must be performed prior to performing a PCR step [2]. Such pre-PCR enrichment processes represent noticeable advantages. Applying them to genomic DNA can circumvent inevitable PCR errors during a pre-amplification step. Moreover, following the genotypic selection step, they can be combined with diverse endpoint detection methods. In order to boost sensitivity even further using RFLP-PCR, a preliminary RE digestion was applied directly to genomic DNA followed by PCR and a second RE digestion [32]. When optimized, this process enriches the gene of interest and subsequently improves the limit of detection (LOD) down to 10^−8^ mutant allelic frequency. In the first reported application of this technique [32], mutagen-exposed human cells were assessed for *TP53* gene hotspot mutations. Following the specific digestion process and PCR, the mutation-containing amplicons were cloned and detected by specific probes.

To partly overcome the major limitation of RE application to solely mutations located within their recognition site, an approach was employed to generate an artificial enzymatic recognition site during PCR synthesis. This method, Artificial Introduction of a Restriction Site (AIRS) employs a modified primer to change one or more nucleotides for the purpose of creating RE recognition sites for WT sequences, but not for mutated sequences [3], thus leading to removal of WT sequences after application of RE on the resulting PCR product. This technique raises the RE limitations and can detect mutations even in the absence of natural restriction sites, at the cost of performing an extra PCR reaction prior to RE digestion, which has the potential for introduction of PCR errors.

In 1998, a thermostable restriction enzyme PCR *BstNI* was employed during a PCR reaction to enable mutated DNA enrichment in a closed tube format. This one-step approach, Restriction Endonuclease-Mediated Selective (REMS)-PCR, was shown to detect effectively *KRAS* mutations in fresh/paraffin-embedded colorectal tumors [4]. The simultaneous activity of *BstNI* endonuclease and *Taq* polymerase, along with employing three sets of primers, results in selective amplification of mutant DNA strands. The PCR products are then visualized via gel electrophoresis. The primer combinations used during REMS-PCR provide information on whether a mutation is present, and whether the restriction enzyme and the polymerase are both active during the reaction. REMS-PCR benefits from the thermostability of *BstNI* to reduce sample handling and to bypass separate enzyme digestion steps, thus shortening the process and reducing the probability for sample cross-contamination. Like RSM and RLFP, this method provides convenience and speed, but the limit of detection LOD is dependent on the ability of the enzyme for complete depletion of mutated strands.

An inverse approach, Amplification via Primer Ligation at the Mutation (APRIL-ATM), detects mutations creating a new enzymatic restriction site. Unlike RFLP, APRIL-ATM does not use the endonuclease activity to eliminate WT alleles, but to digest and ligate mutated alleles to an oligonucleotide tail that acts as primer in a PCR reaction [5]. This method applies an initial pre-amplification to the targeted sequence. Following dephosphorylation, an oligonucleotide is ligated uniquely at the digested position which contains phosphorylated termini. A second round PCR is then performed using the ligated oligonucleotide as a primer to amplify the mutated DNA, and a third PCR is also applied prior to visualizing PCR amplicons on an agarose gel. APRIL-ATM was used for mutation detection in both genomic DNA and cDNA samples [5]. Comparing classical RFLP-PCR with APRIL-ATM, the latter overcomes problems associated with incomplete RFLP enzymatic digestion since only successful enzymatic digestion reveals a mutation. On the other hand, the three PCR steps make APRIL-ATM cumbersome and increase the risk of contamination. Another RE-based technique, FLAG assay (FLuorescent Amplicon Generation), can provide real-time PCR signal generation coupled with mutation enrichment and detection during PCR, using a highly thermostable endonuclease (*PspGI*) [6]. FLAG applies primers doubly labeled with a quencher and a fluorophore that carry a tag sequence containing a *PspGI* recognition site. During primer extension, the tag sequence becomes double-stranded, cleaved by *PspGI*, separating the fluorophore and quencher and generating a detectable signal. Besides real-time mutation enrichment and detection, FLAG is also able to simultaneously genotype mutant alleles using PNA (peptide nucleic acids) probes. This technique could detect and genotype *KRAS* codon 12 mutations in a closed-tube reaction with an LOD of 10^−3^ mutant allelic frequency. The PNA-mediated FLAG approach could detect and genotype low-abundance somatic mutations within a large excess of WT DNA through a single-step process for both tumor biopsies and fluid specimens. Of note, FLAG cannot be applied for targeted regions containing natural *PspGI* recognition sites.

Using bead-based hybrid capture of targeted sequences in combination with the highly discriminatory feature of REs for mutant alleles led to development of Random Mutation Capture (RMC) for detecting random mutations in cancer genomes. This technique integrates bead-based capturing of DNA targets with RE-mediated enrichment [7,33]. Exhaustive digestion of genomic DNA by REs that do not target the studied sequence is followed by a hybridization step with uracil-containing biotinylated probes for enriching the desired sequences. This preliminary digestion boosts the efficiency of depleting WT sequences in the subsequent main RE digestion step. The captured targets are purified by streptavidin magnetic beads and incubated with a specific RE to cleave WT strands but leave intact sequences harboring sites modified due to a mutation. Finally, the undigested strands can be detected by real-time PCR using flanking primers for the RE recognition site. RMC provides a sensitive and quantitative detection approach for multiple mutations. Combining RE digestion with bead-based target enrichment leads to a high sensitivity of ~10^−6^% VAF for detecting minority mutant alleles [7,33]. RMC has the potential to be applied in a multiplexed setting, but it is ultimately limited by the completeness of enzymatic digestion. An enzymatic approach applied to detection of random mutations that overcomes this issue employs inverse PCR, RFLP, and dHPLC for genome-wide mutation scanning. Inverse PCR-based Amplified RFLP (iFLP) applies genomic circularization on fragmented genomic DNA followed by RE digestion on circularized DNA [8]. Sequences with mutations generating a new RE recognition site are cut and converted into linear DNA which can be ligated to generic linkers and amplified. When combined with dHPLC, this approach could screen unknown mutations down to 10^−5^ variant allelic frequency with minimal false positives, since only successfully digested circles could be amplified and detected. iFLP could enrich multiple mutations in colorectal tumor samples to detect unknown random mutations generated by a deficient mismatch repair system. While sensitive and highly specific, iFLP is laborious and detects a limited number of all possible mutations.

Overall, RE-based detection methods render highly selective, available, and low-cost approaches, some of which can be performed directly on genomic DNA without a pre-amplification step. The main drawbacks include the limited number of mutations that can be detected, the possibility of false-positive results due to incomplete enzymatic digestion, and the demanding optimization of certain assay formats.

## 3. Mismatch Recognition Enzymes

Mismatch recognition enzymes are involved in the DNA mismatch repair (MMR) pathway and are evolutionary highly conserved. The MMR pathway in bacteria and mammalian cells can identify and correct base–base mis-pairs and indels. Mismatch-detecting enzymes have been utilized to enrich minority known or unknown mutations in molecular diagnostics. In 1995, immobilized bacterial MutS protein was applied to enrich PCR amplicons containing unknown mutations. MutS recognizes heteroduplex DNA formed via denaturation and cross-hybridization of PCR products containing WT and mutated alleles [9]. MutS was able to detect most mis-paired bases, except C:C, and indels [9]. Following MutS-aided mutation enrichment and amplification, the nucleotide change could be identified via Sanger sequencing. While the immobilized-MutS approach was the first attempt to enzymatic detection of mismatches, the technique is cumbersome due to the required bead-washing steps that may also cause poor reproducibility. MutY is another bacterial-origin enzyme employed to detect mutations via mismatches formed by cross-hybridizing WT DNA with DNA carrying G > T mutations [34]. Subsequently, an approach using MutY or thymine-DNA glycosylase (TDG) which cleave G > A and T > G mismatches, respectively, coupled with ligation-mediated PCR (LM-PCR) was described [10]. In this approach, cross-hybridized mutant/WT DNA duplexes undergo de-phosphorylation to eliminate phosphorylated DNA ends. Next, TDG/MutY enzymes are applied to remove mismatched thymidine or adenine, thereby generating apurinic/apyrimidinic sites (AP sites) that are converted to 5′-phosphate-containing strand breaks via heating. Following oligonucleotide ligation at the 5′P-containing termini, LM-PCR is performed to exponentially amplify selectively the mutation-containing strands. The MutY/TDG/LM-PCR approach is amenable to multiplexing enrichment of unknown mutations. Moreover, the downstream detection can be adapted to bead-based capturing by using biotinylated aldehyde reactive probes [35]. The technique can enrich MutY/TDG-recognized nucleotide mismatches with a moderate selectivity with LOD of ~1% VAF but cannot enrich small deletions, A > T or G > C changes. Endonuclease V is another mismatch recognition enzyme that can identify and cleave heteroduplex DNA one base downstream from the mismatch position [11]. In 2004, a high-sensitivity mutation scanning assay employed Endo V and ligase-based proofreading in a single-step process. The real-time proofreading leads to a remarkable decrease in background cleavage and boosts the sensitivity because DNA ligase preferentially fills in background nicks in perfectly matched regions, but not near mismatched positions. Then, internally labeled primers amplify the cleaved fragments with a 5′-terminus, thereby detecting the mutations. The EndoV/ligase mutation scanning assay demonstrated a variable sensitivity depending on mutation context, with typical LODs of 1–2% VAF for *KRAS* and *TP53* mutations, respectively. T4 endonuclease VII (endo VII) has also been used for cleavage at mismatches [12]. 

The described mismatch recognition-based methods can efficiently detect a fraction of all possible mutations. An alternate method, s-RT-MELT, uses CEL I (Surveyor™) enzyme [36] that shows excellent selectivity and detects all point mutations down to 10^−2^ allele frequency [13]. At first, targets are amplified by region-specific primer pairs containing a high melting domain (GC-clamp) at the 5′ end of the forward primer and an M13 tail at 5′ end of the reverse primer. After cross-hybridization of WT/mutated PCR amplicons and formation of mismatches at mutation positions, Surveyor™ is employed to generate double strand breaks at mismatches. Following digestion, terminal deoxynucleotidyl transferase (TdT) is used to add an oligonucleotide tail to 3′ OH ends created by the digestion. Subsequently, real-time PCR at a reduced denaturation temperature selectively amplifies only the mutation-containing fragments since undigested fragments carry a GC-clamp that does not denature at a reduced temperature. Utilizing real-time PCR coupled with melting curve analysis renders a closed tube approach for unknown mutation screening in clinical samples. PCR amplicons containing mutation-positive strands could be purified and sequenced using primers targeting the M13 tail. The multiplex single-tube s-RT-MELT was used for parallel scanning of mutations in *TP53* exons 5–9. The results demonstrated that the s-RT-MELT approach retains the sensitivity for enriching all types of low-level unknown mutations in a multiplexed fashion and eliminates the requirement for DNA-size-dependent methods such as gel electrophoresis and dHPLC. T7 Endonuclease 1 (T7E1) can also discriminate between homoduplex and heteroduplex dsDNAs. Although it detects and cleaves deletion-induced heteroduplexes more efficiently in comparison with single base-induced heteroduplexes. In the original description of the T7E1 method in 1995, cross-hybridized PCR products underwent T7E1 digestion and then was assessed by gel electrophoresis [14]. A comparative study demonstrated that T7E1 is advantageous for detecting heteroduplexes containing small deletions, while Surveyor performs better in recognizing single nucleotide mismatches [37]. Overall, mismatch-detecting enzymes enable facile enrichment of most or all unknown point mutations, independent of sequence context, and are amenable to multiplexed approaches. However, the selectivity and efficiency of these enzymes are inferior to RE-based mutation enrichment methods.

## 4. Oligonucleotide-Guided Enzymes 

Over the last decade, several mutation enrichment and detection methods have been developed in which synthetic oligonucleotides are used to guide mismatch-sensitive enzymes towards interrogated DNA targets [38]. A broad category of these methods belongs to two endonuclease families, including CRISPR/Cas and Argonaute. The two methods share similar workflows yet have a distinct difference. The enzymatic activity of Cas endonucleases is dependent on the existence of a PAM site within the targeted sequence. PAM is a specific, three-base nucleotide sequence motif located downstream of the region identified by Cas [39]. In contrast, Argonaute (Ago) endonucleases do not require a specific motif for activity, thereby bypassing a limitation [40].

The dependence of Cas enzyme activity on the presence of a PAM site for Cas endonucleases offers a way to identify mutations that alter PAM sites. CRISPR/Cas-based techniques use guide RNAs (gRNAs) coupled with Cas to hybridize to targeted sequences. If gRNA and target sequence form a PAM-containing fully matched hybrid for WT samples, Cas can cleave the targeted double stranded DNA. Thus, similar to the prior discussed RE-based enrichment methods, the ability of CRISPR/Cas system to accurately distinguish between a PAM-containing sequence and a mutant PAM could be employed for enriching rare mutations. The Depletion of Abundant Sequences by Hybridization, DASH (Figure 2), employs CRISPR-Cas9 complex coupled with gRNAs against PAM-containing WT sequence to deplete them in next-generation sequencing libraries or PCR amplicon pools. This process results in enhancing the yield of mutation sequencing to enable a detection limit LOD ~0.1% variant allelic frequency in an affordable approach [15]. DASH was able to increase the percentage of mutant allele *KRAS* G12D in PCR amplicons derived from tumors from 0.1% to 6%. The *KRAS* G12D mutation alters the PAM sequence such that Cas-gRNA complex cannot create a double-strand break effectively for mutated sequences, thus allowing subsequent mutant amplification. The same approach has also been applied during CRISPR-mediated Ultrasensitive detection of Target DNA-PCR (CUT-PCR) with enhanced sensitivity and efficiency [16]. This method also selectively removes high-abundance WT sequences using gRNAs specific to PAM-containing WT DNA. A difference between DASH and CUT-PCR is that CUT-PCR can be applied directly on gDNA prior to PCR, while DASH is mostly applied on PCR-amplified products. Furthermore, in CUT-PCR—besides Cas9 endonuclease—a second enzyme Cpf1 with a different PAM sequence is used. This dual cleavage strategy leads to encompassing nearly 80% of cancer-related substitutions included in COSMIC database [16]. In another recent variation of the same principle, isothermal recombinase polymerase amplification (RPA) was used along with Cas9-mediated WT sequence cleavage [17]. This method, called Programmable Enzyme-Assisted Selective Exponential Amplification (PASEA), can increase the variant allele frequency of mutant alleles up to 800-fold. In the PASEA approach, WT and mutant alleles are amplified concurrently via RPA, while Cas9-gRNA complex selectively degrades WT strands, thereby enriching mutant alleles during isothermal amplification. PASEA shows an unprecedented mutation enrichment; however, its versatile application is limited by the necessity of a PAM motif close to the targeted mutations. In an inverse approach, instead of cleaving WT DNA, deactivated Cas9 enzymes are used to bind mutated DNA via specific gRNAs [18]. In this approach, mutant-specific gRNAs guide a modified version of Cas9 enzymes that can bind but not digest targeted mutations, which subsequently become tagged with polyhistidine (His-tag). Subsequently, mutant strands are separated by immunomagnetic beads. This method could concurrently enrich three hotspot *EGFR* mutations in tumor samples from non-small cell lung cancer (NSCLC) patients up to 20-fold and detect them by qPCR. Recently, CRISPR/Cas9 has been applied in combination with electrochemiluminescence (ECL) technology to enable sensitive single-base-specific DNA detection. For this purpose, gRNAs carrying a 30-nucleotide extension region are used to attach the ECL probe tag and create a ‘light-switch’ molecule. The target sequence is amplified by biotin-tagged forward primers, then the gRNA-Cas9-probe complex hybridized to mutant strands can be captured by Streptavidin-magnetic beads. The integration of Cas9 with an electrochemiluminescence probe (Cas9-ECL probe) can distinguish *EGFR* L858R mutant with a 0.01% VAF from the WT [19]. Since this method is adjustable with the commercial ECL immunoassay analyzer, the endpoint detection process can be simplified in most clinical settings.

Cas9 endonuclease is not the only CRISPR effector enzyme used for mutation enrichment. Two similar platforms, Specific High Sensitivity Enzymatic Reporter UnLOCKing (SHERLOCK) and one-HOur Low-cost Multipurpose Highly Efficient System (HOLMES), employ Cas13a and Cas12a endonucleases, respectively [20,21]. Both these techniques benefit from a specific digestion pattern of Cas13a and Cas12a named “collateral effect” [41]. The collateral effect enables them to cleave single-stranded guide RNA/DNA in a nuclease-like pattern. By using single stranded probes labeled by quencher and fluorescent, a fluorescent signal is generated via the collateral effect, upon forming a fully matched Cas/gRNA/target complex. Recently, Cas12a mediated Surface Enhanced Raman Scattering (SERS) was combined with lateral flow assay (LFA) to render a sensitive pre-PCR enrichment method based on CRISPR-Cas system [22]. Using CRISPR-Cas12a mediated SERS LFA, point mutations as low as 0.01% can be identified. Although all CRISPR/Cas-based enrichments methods require a mutation that is adjacent to a PAM site to operate, the utility of these methods could potentially be expanded by modifying their required PAM sequence through genetic engineering.

In contrast to Cas endonucleases, the enzymatic activity of Ago family is not dependent on the existence of a sequence motif. This feature makes Ago-based methods more versatile for mutation detection. The PfAgo-mediated Nucleic acid Detection method (PAND) is the first Ago-based process developed to detect mutant alleles and has an LOD ~0.1% VAF [23]. PAND employs *Pyrococcus furiosus* Argonaute (PfAgo) for a sequential two-stage cleavage process. In the first round, the enzyme cleaves the hybridized three input-gDNAs such that a new gDNA (ngDNA) is formed in which the mutant allele exists. The ngDNA is a 16-nucleotide single-stranded DNA (ssDNA) that triggers a second round of PfAgo activity to cleave fully matched molecular beacons. The synthetic molecular beacon is fully matched with mutant alleles and can be detected and cleaved by ngDNA-PfAgo complex. By this means, mutations at a targeted sequence are detected by fluorescence emission. The A-Star technique (Figure 2) also uses the thermostable PfAgo enzyme. However, in A-Star, gDNAs are designed for attaching to WT strands [24]. A-Star leads to a strong decline in unwanted WT amplicons during PCR, resulting in ~75-fold mutation enrichment. Another member of Ago endonuclease family from *Thermus thermophilus*, *TtAgo*, has also been applied for eliminating WT templates enabling a ~60-fold enrichment [25]. In this method, Nucleic Acid enrichment Via DNA Guided Argonaute from *Thermus thermophilus* (NAVIGATER, Figure 3), the enzymatic process and PCR are performed in successive steps to improve sensitivity. The TtAgo enzyme is thermostable, and its activity is not significantly impaired by high temperatures. The 15–16 nt gDNAs hybridize to both sense and antisense strands of WT sequences at ~80 °C during a one-hour incubation. Mismatches in gDNA/target hybrids formed due to mutations impede TtAgo activity. Noticeably, during NAVIGATER there is no distinct denaturation step, and dsDNAs unwind partially during the high temperature (~80 °C) and long incubation time applied. Another feature of NAVIGATER is that it can also be performed in a pre-PCR setting, although this can lead to a reduced mutation-enrichment efficiency.

Besides Cas and Ago endonucleases, additional oligonucleotide-guided enzymes have also been applied for mutation enrichment. The apurinic/apyrimidinic-probe-based endonuclease IV signal amplification system (APESA) uses endo IV enzyme and apurinic/apyrimidinic probes (AP-probe) to detect point mutations [26]. The sequence of the AP-probes matches mutation-containing sequences and is dually labeled by the quencher and fluorophore. Since endo IV recognizes AP sites within double stranded DNA, it cleaves the hybridized probes at optimum Tm. Consequently, the fluorophore fragment is separated from the quencher, resulting in fluorescence signals. A remarkable point is that the mutant template will remain intact so that it can hybridize with another AP-probe and intensify the signals. In 2022, using endo IV and asymmetric PCR, mutant alleles with VAF ~0.01% could be enriched and detected through the endonuclease IV-mediated substrate structure allosteric (IVME) method [27]. This assay benefits from the property of endo IV to preferentially hydrolyze AP sites that have a fully matched nucleotide at -1 position, when the AP site’s position is set as 0, in truncated dsDNAs. IVME employs mutant-specific probes containing an AP site next to the mutant nucleotide. The probes have a flap structure at the 3′ terminals to prevent an asymmetric PCR reaction. The probe-mutant hybrids are preferred substrates of Endo IV in comparison with probe-WT hybrids in which there is a mismatch downstream of the AP site. Endo IV also generates a 3′-OH terminus at the single-stranded cleavage site which is used by DNA polymerase. Upon cleaving the AP site of probe-mutant hybrids, the emerging 3′-OH is extended by DNA polymerase. Finally, the dsDNAs formed from the previous step undergo asymmetric PCR that provides ssDNAs compatible with different detection methodologies. IVME was used to enrich *EGFR* T790M mutation in cfDNA of lung cancer patients, followed by Sanger sequencing.

Thermostable duplex-specific nuclease (DSN) is also employed to develop a multiplex enrichment method that is directly applied to genomic DNA, Figure 4 [28]. Nuclease-Assisted Minor-Allele enrichment using Probe Overlap (NaME-PrO) utilizes DSN to cleave fully matched double stranded DNA, with almost no sequence dependence, while sparing mismatched DNA. Overlapping oligonucleotide probes are designed such that fully bind both sense and antisense WT sequences at the interrogated position. NaME-PrO is applied directly to genomic DNA, prior to PCR, and provides multi-targeted enrichment that could be followed by various types of endpoint detection methods. NaME-PrO can be applied both on intact DNA or, or on DNA obtained from liquid biopsies or FFPE tissues, since the degree of fragmentation [42] does not significantly affect the result [43]. Additionally, the NaME-PrO process was also adapted for identification of microsatellite instability (MSI). MSI-NaME-PrO could trace MSI in a pre-PCR setting by removing unaltered micro-satellites [44]. Another modification named methylation-specific NaME-PrO (MS-NaME) has been developed for detecting aberrant methylation signatures [45]. Following sodium bisulfite treatment, depending on whether methylated or unmethylated sequences are desired for enrichment, either U-probes targeting uracil-containing sequences or M-probes targeting ^5m^C-containing DNA are used to guide DSN. The former results in degrading unmethylated DNA and the latter is used for digesting methylated DNA. NaME-PrO, MS-NaME-PrO, and MSI-NaME-PrO have been implemented in multiplexed formats for simultaneous enrichment of 55, 177, and several thousand targets, respectively [44,45,46].

Collectively, mutation enrichment methods based on oligonucleotide-guided enzymes pose robust and versatile platforms in which some drawbacks of traditional techniques are circumvented, such as limited types of mutations identified, futile binding events, or off-target cleavages. Moreover, a noticeable feature is their potential for running a multiplex enrichment process.

## 5. Applications in Liquid Biopsy

The importance of detecting low-level mutations in clinical fluids such as circulating tumor cells (CTC) or circulating-free DNA (cfDNA) has been demonstrated repeatedly in recent years. In early work, using BEAM-ing, Misale et al. [47] demonstrated that *KRAS* resistance-mutations at abundance levels of ~0.1% VAF appear early-on in circulating DNA, thus potentially enabling timely identification of patients that do not respond to targeted therapy. Using COLD-PCR-based mutation enrichment [48,49,50,51], it was demonstrated that NRAS mutations are an independent prognostic factor for myelodysplastic syndrome at allelic frequencies ≥ 0.5% [52]. Overall, liquid biopsies have made significant strides in cancer medicine over the last few years, with applications spanning early detection of minimal residual disease, MRD [53,54,55,56,57], early cancer detection [58,59,60,61,62], assessment of therapy effectiveness [63,64] and drug-resistance [65], as well as monitoring tumor-dynamics [66]. The time course of tumor-circulating DNA in plasma following initiation of therapy [67] can be prognostic [68], and often an initial ctDNA rise is followed by a ctDNA decrease [69]. The ctDNA time-course was originally demonstrated during uniform, external beam radiation therapy [70], while the ctDNA dynamics present in tumor brachytherapy and other types of radiation exposure that deliver highly non-uniform radiation-induced, lethal DNA damage [71,72] to nearby cells [73,74,75,76] remain to be explored. 

Despite these advances, working with circulating tumor cells (CTCs) and circulating free DNA (cfDNA) still presents technical challenges. For example, profiling somatic mutations in circulating DNA has been difficult because of their extremely low frequency in the presence of high excess WT alleles. To this end, the availability of sensitive mutation enrichment and detection technologies is important for realizing the clinical utility of liquid biopsies [77]. NGS-based approaches for accurate enumeration of mutations as low as 0.01–0.001% have been reported [51,74,75,76], but NGS still requires advanced instrumentation, resources, and time. When a limited number of somatic mutations is interrogated, genotyping approaches—including enzymatic enrichment methods—provide a practical solution more appropriate for low-resource or non-specialized clinical settings.

Indeed, several of the recently developed enzymatic methods have been applied for liquid biopsy applications. NaME-PrO-based enrichment has been applied for enriching PIK3CA hotspot mutations in cfDNA samples from breast cancer (BC) patients [78]. The NAVIGATER method has also been used for enriching and detecting KRAS G12D/V/R mutations in cfDNA from patients with pancreatic cancer [25]. The results demonstrated that NAVIGATER has a selectivity of <0.2% VAF for cfDNA analysis. Recently, a CRISPR/Cas9-based mutant enrichment technique revealed a 93.9% sensitivity and 100% specificity in detecting *EGFR* T790M mutation in cfDNA from patients with NSCLC [79]. In colorectal cancer patients, CUT-PCR method could enrich five informative *KRAS* mutations in cfDNA (c.34G4T, c.35G4T, c.35G4C, c.34G4C, and c.35G4A) by a common gDNA at the same time. The results indicated that the mutant frequency increases up to 600-fold [16]. In 2022, CRISPR system combined with post-PCR cfDNA (CRISPR-CPPC), using biotinylated gRNA and Cas9 enzyme, could enrich *EGFR* exon19 deletion up to 1000-fold in cfDNA samples from NSCLC patients [79]. This method comprises the pre-PCR step, post-PCR step coupled CRISPR/Cas9 activity, and magnetic-based enrichment. The endpoint detection method in this study was ddPCR, although the authors pointed out that CRISPR-CPPC could be followed by a variety of detection platforms.

The amount of starting DNA for application of each method can vary depending on whether there is an initial pre-amplification step or not. Methods that start with PCR can operate with just sub-nanogram amounts of starting material while methods starting with enzymatic digestion require more DNA, typically a minimum of 1–10 ng. On the other hand, pre-amplification increases the risk for PCR-introduced errors [80], especially for those enzymatic approaches that enable mutation scanning (Figure 1). Liquid biopsies that contain small amounts of starting DNA require several PCR cycles for amplification and can be particularly susceptible to PCR errors. Strategies to overcome polymerase introduced errors have been developed, based on a principle described by Kaur and Makrigiorgos in 2003 [80,81]. While polymerase errors may occur only on a single-sense or anti-sense DNA strand, genuine mutations are present on both DNA strands on single DNA molecules. Accordingly, it is possible to distinguish polymerase-introduced errors from genuine mutations if one tracks nucleotide changes on both sense and anti-sense strands in individual DNA molecules [80,81]. This principle was subsequently incorporated in duplex sequencing during NGS that utilizes unique molecular identifiers (duplex-UMI) to track mutations on both DNA strands of single duplexes [82]. While NGS-based approaches can reduce the effect of PCR errors by using duplex-UMIs, non-NGS applications including the enzymatic mutation detection methods are still susceptible to PCR errors when a genotypic selection step is applied after or during PCR amplification. Standard precautions to minimize errors include the use of proof-reading polymerases and inclusion of experimental replicas to distinguish reproducibly detectable mutations from randomly occurring DNA errors. Algorithms to minimize the effect of PCR errors have been developed for digital PCR [83].

## 6. Conclusions

Enzymatic-based mutation enrichment methods have evolved through the years from the traditional approaches using restriction endonucleases that detect a minority of mutations to newer methods using a variety of different enzymes that can detect all mutations on multiple targets simultaneously. In recent years, employing endonucleases with broader catalytic capabilities has rendered sensitive, multiplex, and affordable enrichment processes. When it comes to liquid biopsy, enzymatic mutation detection provides a practical approach in view of its simplicity and availability in most clinical laboratories. Overall, recent advances in enzymatic approaches for mutation detection promise a broad application for detecting clinically relevant mutations, with applications in several fields including cancer management.

## Figures and Tables

**Figure 1 ijms-24-00923-f001:**
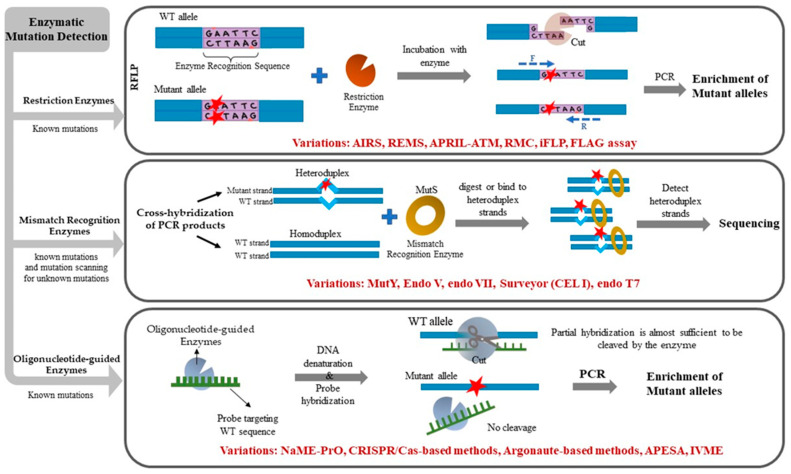
Schematic figure illustrating the concept behind the three different approaches of enzymatic mutation detection including restriction enzymes, mismatch recognition enzymes, and oligonucleotide-guided enzymes, including subsequent variations and enhancements for each method.

**Figure 2 ijms-24-00923-f002:**
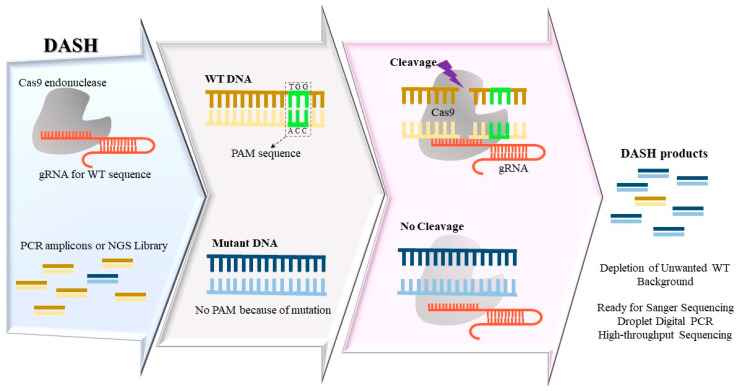
Depletion of Abundant Sequences by Hybridization (DASH) method removes excessive wide-type (WT) DNA in PCR amplicons using guide RNA (gRNA) against PAM-containing WT sequences, in conjunction with Cas9 enzyme. When a mutation alters the PAM sequence, the Cas9-gRNA complex cannot create a double-strand break efficiently, thus the intact mutant strands are amplified selectively and are detected via subsequent detection methods.

**Figure 3 ijms-24-00923-f003:**
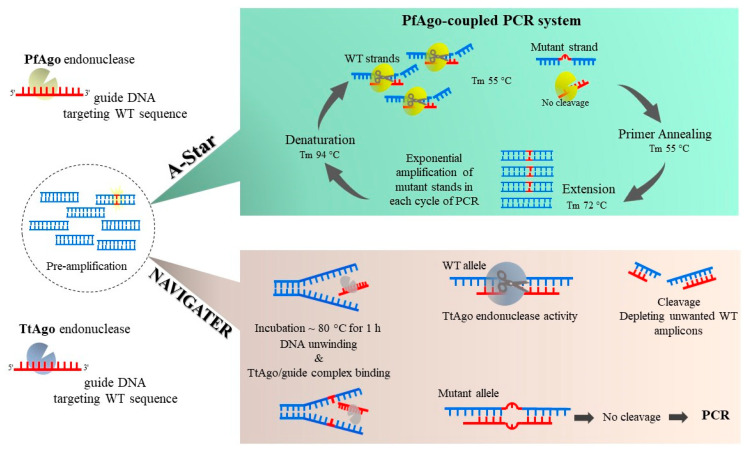
A-Star and NAVIGATER methods enable enzymatic mutation detection that does not require the presence of a specific motif site near the mutation site. The A-Star method uses a thermostable PfAgo enzyme, that makes a complex with guide DNA (gDNA) that matches WT strand sequence. PfAgo-coupled PCR reactions are performed to cleave WT strands repeatedly during PCR cycling and subsequently increase mutant strands exponentially. In the Nucleic Acid enrichment VIa DNA-Guided Argonaute enzyme (NAVIGATER) method, from *Thermus thermophilus*, WT alleles are detected and cleaved using *Tt*Ago and a gDNA to bind complementary WT strands. Following an initial pre-amplification step, the amplicons unwind during incubation at ~80 °C and provide the chance for *Tt*Ago/guide complexes to attach to WT alleles. *Tt*Ago detects fully matched WT/gDNA hybrids while sparing mismatches caused by mutations. A subsequent PCR reaction amplifies mutated alleles preferentially, and these are detected via subsequent detection methods.

**Figure 4 ijms-24-00923-f004:**
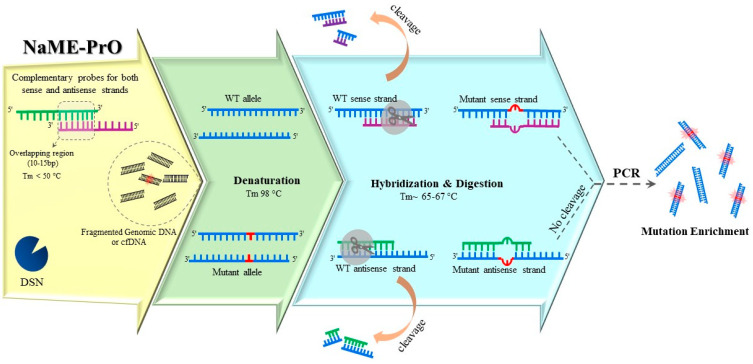
Nuclease-assisted minor-allele enrichment using probe-overlap (NaME-PrO) is applied directly to unamplified genomic DNA, thus reducing the influence of polymerase misincorporations. Using duplex-specific nuclease (DSN) along with overlapping WT-specific probes results in digesting WT alleles preferentially. Following DNA denaturation, the temperature is lowered up to 65 °C, allowing the probe hybridization and catalytic activity of DSN. Hybrids with mismatches due to any mutation within the overlap region of sense and antisense probes escape DSN digestion. The resulting DNA is PCR-amplified, resulting in mutation enriched products.

**Table 1 ijms-24-00923-t001:** Enzymatic mutation enrichment and detection techniques. VAF: Variant Allele Frequency.

Enzymatic Methods	Application of Genotypic Selection	Type of Mutation	Sensitivity (%VAF)	Year (Reference)
Restriction Enzymes				
RSM-PCR ^(1)^	Pre-PCR	Known point mutation	0.0001–0.1	1990 [1]
RFLP-PCR ^(2)^	Pre-PCR	Known point mutation	0.0001–0.1	2004 [2]
AIRS-RFLP ^(3)^	Post-PCR	Known point mutation	0.01–0.1	1989 [3]
REMS ^(4)^	Pre/Post-PCR	Known point mutation	0.01	1998 [4]
APRIL-ATM ^(5)^	Post-PCR	Known point mutation	0.01–0.1	2002 [5]
FLAG assay ^(6)^	Pre-PCR	Known point mutation	0.1–1	2007 [6]
RMC ^(7)^	Pre-PCR	Known point mutation	0.0001–0.1	2005 [7]
iFLP ^(8)^	Pre-PCR	Unknown point mutation	0.001–0.1	2004 [8]
Mismatch Recognition Enzymes				
MutS	Post-PCR	Unknown point mutation	1–5	1995 [9]
MutY/TDG-LM-PCR	Post-PCR	Unknown point mutation	1–5	2002 [10]
Endo V	Post-PCR	Unknown point mutation	1–5	2004 [11]
Endo VII	Post-PCR	Unknown point mutation	1–5	1995 [12]
sRT-MELT	Post-PCR	Unknown point mutation	1–5	2007 [13]
T7E1	Post-PCR	Unknown deletion/point mutation	1–5	1995 [14]
Oligonucleotide-Guided Enzymes				
DASH ^(9)^	Post-PCR	Known point mutation	0.1	2016 [15]
CUT-PCR	Pre-PCR	Known point mutation	0.01	2022 [16]
PASEA ^(10)^	Post-PCR	Known point mutation	0.01	2021 [17]
dCas9-based minor-allele enrichment	Post-PCR	Known point mutation	0.1	2018 [18]
Cas9-ECL probe ^(11)^	Pre-PCR	Known point mutation	0.01	2022 [19]
SHERLOCK ^(12)^	Post-PCR	Known point mutation	0.1	2017 [20]
HOLMES ^(13)^	Post-PCR	Known point mutation	0.1	2018 [21]
CRISPR-Cas12a mediated SERS-LFA ^(14)^	Pre-PCR	Known point mutation	0.01	2022 [22]
PAND ^(15)^	Post-PCR	Known point mutation	0.1	2019 [23]
A-Star	Post-PCR	Known point mutation	0.01–0.1	2021 [24]
NAVIGATER ^(16)^	Pre-PCR	Known point mutation	0.01–0.1	2020 [25]
APESA ^(17)^	Post-PCR	Known point mutation	0.01–0.1	2012 [26]
IVME ^(18)^	Post-PCR	Unknown point mutation	0.01–0.1	2022 [27]
NaME-PrO ^(19)^	Pre-PCR	Known point mutation	0.01–0.1	2016 [28]

^(1)^ Restriction-site mutation; ^(2)^ Restriction Fragment Length Polymorphism-PCR; ^(3)^ Artificial introduction of a restriction site; ^(4)^ Restriction Endonuclease-Mediated Selective; ^(5)^ Amplification via Primer Ligation at the Mutation; ^(6)^ FLuorescent Amplicon Generation; ^(7)^ Random Mutation Capture; ^(8)^ Inverse PCR-based Amplified RFLP; ^(9)^ Depletion of Abundant Sequences by Hybridization; ^(10)^ Programmable Enzyme-Assisted Selective Exponential Amplification; ^(11)^ *Cas9*-electrochemiluminescence probe; ^(12)^ Specific High Sensitivity Enzymatic Reporter UnLOCKing; ^(13)^ One-HOur Low-cost Multipurpose highly Efficient System; ^(14)^ *Cas12a* mediated Surface Enhanced Raman Scattering-lateral flow assay; ^(15)^ *PfAgo*-mediated Nucleic acid Detection method; ^(16)^ Nucleic Acid enrichment Via DNA Guided Argonaute from *Thermus thermophilus*; ^(17)^ Apurinic/apyrimidinic-probe-based *endonuclease IV* signal amplification system; ^(18)^ *Endonuclease IV*-mediated substrate structure allosteric; ^(19)^ Nuclease assisted Minor-Allele enrichment using Probe Overlap.

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
