# Peer review of "Enzymatic Methods for Mutation Detection in Cancer Samples and Liquid Biopsies"

_ijms, 2023, doi:10.3390/ijms24020923_

Round 1
Reviewer 1 Report
The manuscript is well written and could be helpful for the readers. The authors critically reviewed many technologies used for mutation detection.
The manuscript summarizes old and new techniques exploiting enzymatic mutation detection methodologies.The manuscript shows the potential of modern techniques based on CRISPR - Cas and Argonaute proteins, which allows testing a larger number of mutations using the NGS method. Overall, the above methodology is very important for the wider introduction of liquid biopsy based on NGS analysis of large panels.
Please prepare a scheme figure for each mentioned technology in chapters 2 and 3 to easily follow the reader's whole article.
Author Response
Answers to the comments of the reviewers ijms-2092506
We thank the reviewers for the positive comments on our manuscript. We have now incorporated changes in the manuscript that address all comments.
Reviewer 1: Please prepare a scheme figure for each mentioned technology in chapters 2 and 3 to easily follow the reader's whole article.
In view of the recommendation to include a new figure, we have now included Figure 1 that describes the three main principles, how these work, and list the variants of each method described in sections 2 and 3 under the appropriate principle. This should make the text easier to follow.
Reviewer 2: The figures are very helpful to get an overview of specific assays. An additional summary figure showing the different enzymatic methods (RE, MMR, Oligo-guided etc.) schematically (how they work), would enhance this review.
We have now included Figure 1 that lists the main principles and the variations of each method from the main principle, as also recommended by Reviewer 1.
There is no mention of T7 Endonuclease 1, which is used to detect mutations in genome engineering, similar to the Surveyor assay
We have now incorporated the T7E1 method in the text lines 220-227.
Some discussion on the amount of input DNA needed for some of the enzymatic assays would be helpful to readers. Particularly for ctDNA, extensive PCR amplification is presumably required due to limited initial starting material - what is the error rate introduced here, and compare this to the mutation detection limit (whether the test is highly sensitive, but the amount of PCR amplification required would mean the high sensitivity is moot?)
We thank the reviewer for this important comment that enhances our review. We have now added a paragraph at the end of the manuscript lines 452-474 that discusses minimum input DNA needed as starting material, the influence of PCR errors, and methods to overcome them.
Reviewer 2 Report
The authors have done a very good job of summarising a large amount of information in an easily understandable format. It is well structured and written. I have the following comments and suggestions, which should be possible to address through minor revisions:
1. The figures are very helpful to get an overview of specific assays. An additional summary figure showing the different enzymatic methods (RE, MMR, Oligo-guided etc.) schematically (how they work), would enhance this review.
2. There is no mention of T7 Endonuclease 1, which is used to detect mutations in genome engineering, similar to the Surveyor assay (several publications on this, see https://www.ncbi.nlm.nih.gov/pmc/articles/PMC4349094/ for a direct comparison) - while T7E1 may not have been used in the context of cancer mutations, I believe that it has similar utility as the CEL / Surveyor enzyme assay and worth discussing particularly due to its use for CRISPR mutation detection.
3. Some discussion on the amount of input DNA needed for some of the enzymatic assays would be helpful to readers. Particularly for ctDNA, extensive PCR amplification is presumably required due to limited initial starting material - what is the error rate introduced here, and compare this to the mutation detection limit (whether the test is highly sensitive but the amount of PCR amplification required would mean the high sensitivity is moot?)
Author Response

(The authors gave the same response as above.)
